# Changes in the Volatile Profile of Wheat Sourdough Produced with the Addition of Cava Lees

**DOI:** 10.3390/molecules27113588

**Published:** 2022-06-02

**Authors:** Alba Martín-Garcia, Oriol Comas-Basté, Montserrat Riu-Aumatell, Mariluz Latorre-Moratalla, Elvira López-Tamames

**Affiliations:** 1Departament de Nutrició, Ciències de l’Alimentació i Gastronomia, Facultat de Farmàcia i Ciències de l’Alimentació, Campus de l’Alimentació de Torribera, Universitat de Barcelona, Av. Prat de la Riba 171, 08921 Santa Coloma de Gramenet, Spain; albamartin@ub.edu (A.M.-G.); oriolcomas@ub.edu (O.C.-B.); mariluzlatorre@ub.edu (M.L.-M.); e.lopez.tamames@ub.edu (E.L.-T.); 2Institut de Recerca en Nutrició i Seguretat Alimentària (INSA-UB), Universitat de Barcelona, Av. Prat de la Riba 171, 08921 Santa Coloma de Gramenet, Spain; 3Xarxa d’Innovació Alimentària de la Generalitat de Catalunya (XIA), C/Baldiri Reixac 4, 08028 Barcelona, Spain

**Keywords:** sourdough, HS-SPME-GC-MS, volatile compounds, Cava lees, wine by-product

## Abstract

The volatile fraction is of great importance for the organoleptic quality and consumer acceptance of bread. The use of sourdough improves the sensory profile of bread, as well as the addition of new ingredients to the fermentation. Cava lees are a sparkling wine by-product formed of dead microorganisms, tartaric acid, and other inorganic compounds, rich in antioxidant compounds as well as β-glucans and mannoproteins. The aim of this study was to evaluate the effect of different concentrations of Cava lees (0–2% *w*/*w*) on sourdough volatile compounds to re-valorize this by-product of the wine industry. Headspace solid-phase microextraction (HS-SPME) was optimized to study the volatile fractions of sourdoughs. The parameters selected were 60 °C, 15 min of equilibrium, and 30 min of extraction. It was found that the addition of Cava lees resulted in higher concentrations of volatile compounds (alcohols, acids, aldehydes, ketones and esters), with the highest values being reached with the 2% Cava lees. Moreover, Cava lees contributed to aroma due to the compounds usually found in sparkling wine, such as 1-butanol, octanoic acid, benzaldehyde and ethyl hexanoate.

## 1. Introduction

Sourdough is the result of fermenting a mixture of flour and water, and it is traditionally used during bread making as a leavening agent, influencing bread quality [1,2]. This process takes place by the action of the lactic acid bacteria (LAB) and yeasts present in flour, and can occur either by the addition of a starter culture or by spontaneous fermentation. The metabolic activity of the bacteria leads to acidification and flavor formation, improving nutritional and sensory characteristics in addition to increasing microbiologic stability and shelf life [3].

The volatile profile is very significant for the organoleptic quality and consumer acceptance of bread. More than 500 volatile compounds have been reported in bread [4], while sourdough (and sourdough bread) volatiles have been less studied, with less than 200 compounds having been identified [2]. Several research articles have been published in which headspace solid-phase microextraction (HS-SPME) has been used to study the volatile fraction of sourdough [5,6,7,8]. Nonetheless, there is no common base methodology.

Moreover, sourdough bread flavor strongly depends on the fermenting microbiota that produces a range of secondary metabolites, as well as on the enzymatic and autoxidation of flour lipids, and the Maillard reaction [1,2,3]. In addition, several bacteria and yeast strains not only produce desirable volatile compounds, but also release aromatic precursors, and some are able to degrade undesirable compounds [7,9]. In fact, researchers are studying the use of different flours (i.e., chickpea, lentil, bean and hemp) and the addition of new ingredients (i.e., broccoli by-products and brewers’ spent grain) in sourdough formulation that can improve its fermentation and the effect on sourdough and bread characteristics [7,10,11,12,13]. On that account, our research group has been focused on the valorization of wine by-products as new ingredients in bakery products. We found that the use of Cava lees in wheat and rye sourdough promoted the growth and survival of LAB and yeast in spontaneous fermentation [14].

Lees are a residue formed during the ageing process of Cava (Spanish sparkling wine) and consist, mostly, of dead microorganisms (generally *Saccharomyces cerevisiae*), tartaric acid, and other adsorbed compounds [15,16]. They are rich in antioxidant compounds [16,17] as well as dietary fiber from the yeast cell wall that is composed of mannoproteins and branched β-glucans [18,19]. Nowadays, Cava lees are produced at an amount of 300 tons per year, representing 25% of the waste generated by the wine industry [17]. Although some studies have reported that wine lees could acquire an added value due to their composition [18,20,21,22,23], they are actually destined for distillation. Moreover, there is an increasing tendency in the food industry towards reducing food waste and re-valorizing by- and co-products to contribute to a circular economy and sustainable food production [10,13,21,24]. 

The addition of Cava lees in the formulation of sourdough could have an important effect on the fermenting microbiota and, hence, in the volatile profiles of these products. Therefore, the aim of this study was to evaluate the impact of Cava lees on sourdough volatile compounds by an optimized method of HS-SPME-GC-MS.

## 2. Results

The addition of Cava lees to sourdough formulation may change the volatile profiles of such products. Hence, this study focused on the impact of different concentrations of Cava lees on the volatile fraction of wheat sourdough. Since there is no common base methodology for the extraction of volatile compounds in sourdough [5,6,7,8], a previous optimization of the HS-SPME parameters was performed. 

### 2.1. Optimization of Headspace Solid-Phase Microextraction (HS-SPME) Parameters

Figure 1 shows the total number (TN) of volatile compounds and total area (TA) of volatile compounds identified by GC-MS analysis as a result of the modification of the extraction parameters. Figure 2 shows the TN of the different chemical families (acids, alcohols, aldehydes and ketones, and esters) identified.

#### 2.1.1. Effect of Heating Temperature

To evaluate the effect of the heating temperature, four temperatures were selected: 20 °C, 50 °C, 60 °C, and 75 °C. The impact of heating temperature on the extraction of the volatile compounds from sourdough is shown in Figure 1a and Figure 2a. The TN and TA of the compounds increased with temperature (Figure 1a), although there were no significant differences between the TN of compounds extracted between 50 °C (250 ± 12 identified compounds), 60 °C (259 ± 13 identified compounds), and 75 °C (263 ± 12 identified compounds). However, Figure 2a shows that when compounds were separated by chemical families, the TN did not increase with temperature for all of them. Acids, aldehydes and ketones increased with temperature. Alcohols and esters decreased, reaching the maximum performance in the TN of volatiles extracted at 50 °C (67 ± 1.2 and 64 ± 1.0 compounds, respectively), although there were no significant differences (*p* > 0.05) between 50 °C and 60 °C (63 ± 1.2 (alcohols) and 60 ± 1.0 (esters)). Since 60 °C was the temperature at which the number of compounds extracted was higher, it was the selected temperature for the HS-SPME in sourdough.

#### 2.1.2. Effect of Equilibrium Time

For the optimization of the HS-SPME method, three periods of time were assessed for the equilibration of the samples: 10 min, 15 min and 30 min (Figure 1b and Figure 2b). It can be observed that the equilibrium times before extraction did not lead to any significant differences in the TN identified, in general or when separating between chemical families. In view of the cost of time, an equilibrium time of 15 min was chosen as the optimal amount of time sufficient to extract the volatile compounds of the sourdough.

#### 2.1.3. Effect of Extraction Time

Four periods of time were tested for extraction: 20 min, 30 min, 40 min and 50 min. The TN and TA of the volatile compounds extracted depending on extraction time are shown in Figure 1c. Figure 2c shows the TN of each chemical family according to different times of extraction. There was an increase in the TN of volatiles extracted when increasing the extraction time, although there was a decrease in those numbers with 50 min of extraction (Figure 1c and Figure 2c). Therefore, the optimal extraction time was considered to be between 30 min and 40 min, which were the periods of time that showed the maximum number of volatiles identified (268 ± 8.7 and 262 ± 12.7 identified compounds, respectively). When observing the impact of time on each chemical family, the effect was similar on all of them, except for esters, which peaked at 30 min (56 ± 0.7 compounds) and began decreasing at 40 min (46 ± 1.2 compounds). Regarding the other compounds, there were no significant differences between the TN of compounds extracted at 30 min and 40 min.

As shown in Figure 3, at a lower temperature (20 °C) and shorter extraction time (20 min), the TN of components extracted was significantly reduced (33 ± 1.4 compounds) compared to the same temperature with a longer extraction time of 40 min (48 ± 1.8 compounds). Nonetheless, when the extraction time was too long (50 min), the TN of components was also lower (29 ± 1.2 compounds). The same trend was observed for all the other studied temperatures (50 °C, 60 °C and 75 °C). For the selected temperature of 60 °C, the TN increased from 36 ± 0.7 (20 min extraction) to 79 ± 0.9 (30 and 40 min) compounds. However, when the time of extraction was 50 min, the TN compounds identified decreased (65 ± 0.5). Therefore, the extraction time selected was 30 min in order to obtain the highest number of volatiles extracted from each chemical family with the shortest amount of time possible.

### 2.2. Analysis of Volatile Compounds in Different Sourdough Samples

The effect of different percentages of Cava lees on the volatile profile were assessed following the optimized HS-SPME method (60 °C, 15 min, 30 min). The volatile compounds of Cava lees were also analyzed by HS-SPME. During the sourdough fermentation, volatile compounds such as alcohols, acids, aldehydes, ketones, and esters were formed (Table 1).

Overall, the control and 0.5% Cava lees samples showed no significant differences in the concentration of volatile compounds reported, except for esters (*p* < 0.05). As a general rule, with higher amounts of Cava lees added to the sourdough formulation, there was a greater production of volatile compounds, especially alcohols and esters (*p* < 0.05). In fact, Cava lees were characterized by esters (489.38 ± 60.34 mg/kg).

#### 2.2.1. Alcohols

The concentration of alcohols (Table 1) increased with the addition of lees, with values ranging between 158.66 ± 28.81 mg/kg (control) and 923.39 ± 150.02 mg/kg (2% Cava lees). The main alcohols identified in sourdough were 1-butanol, 1-pentanol, 1-hexanol, 1-octen-3-ol, 1-heptanol, 1-octanol, 1-nonanol, and 2-ethylhexanol (Table 2).

Most of the alcohols identified in the sourdough samples increased their concentration with the addition of Cava lees, reaching the highest values at 2% Cava lees (*p* < 0.05). The dominant alcohol quantified was 1-hexanol in sourdoughs formulated with and without Cava lees. Additionally, 1-butanol was the dominant alcohol in sourdoughs with 2% Cava lees (249.65 ± 29.86 mg/kg).

#### 2.2.2. Acids

The total concentration of acids ranged between 386.83 ± 99.92 mg/kg and 1008.03 ± 69.33 mg/kg, depending on the Cava lees percentage in the sourdough. A total of 12 different acids were found in the sourdough samples (Table 3): acetic, butanoic, pentanoic, hexanoic, heptanoic, octanoic, nonanoic, decanoic, benzoic, tetradecanoic, hexadecenoic, and octadecanoic acid.

Although butanoic acid (269.41 ± 9.36 mg/kg) was the most prevalent acid in the control sourdough, its concentration decreased with the addition of Cava lees, and was not detected in the 2% Cava lees sourdough. Pentanoic and heptanoic acid followed the same trend, being identified in the control sourdough but not in the sourdoughs with Cava lees. The opposite occurred to acetic acid, with a lower concentration in the control sourdough (66.08 ± 9.99 mg/kg) that increased with lees, reaching values of 246.10 ± 14.64 mg/kg with the 2% Cava lees. 

#### 2.2.3. Aldehydes and Ketones

The aldehydes identified in sourdough (Table 4) were hexanal, benzaldehyde, nonanal, (E)-2-heptenal, (E)-2-octenal, and decanal. In the control and 0.5% lees sourdoughs, the most prevalent aldehyde was (E)-2-octenal (72.65 ± 3.32 mg/kg and 75.18 ± 5.44 mg/kg, respectively), while in 1% and 2% lees sourdoughs this aldehyde was not detected. Generally, hexanal, nonanal and decanal increased their concentration with the addition of Cava lees, adding 1% to 2% (*w*/*w*). The ketones quantified in this study were acetoin and 2-undecanone, representing 1% of the total volatile fractions in all samples (Table 1). Acetoin was the main ketone identified in all samples.

#### 2.2.4. Esters

Esters were the most prevalent compounds in all types of sourdough (Table 1), especially when Cava lees were added, with values ranging between 373.40 mg/kg in the control sourdough (33% of the total volatile compounds) and 7514.18 mg/kg in the 2% Cava lees sourdough (77% of the total volatile compounds). The esters present in the sourdough (Table 5) were butyl acetate, butyl butyrate, butyl hexanoate, butyl benzoate, ethyl hexanoate, ethyl octanoate, ethyl decanoate, ethyl laurate, ethyl palmitate, hexyl acetate, hexyl butyrate, octyl acetate, and decyl acetate. The most prevalent esters found in the sourdough samples were butyl butyrate (96.76 ± 3.36 mg/kg–689.04 ± 12.35 mg/kg) and ethyl decanoate (80.07 ± 15.41 mg/kg–3,330.26 ± 314.82 mg/kg) in all sourdoughs. Moreover, the sourdoughs with Cava lees presented high concentrations of ethyl octanoate (170.91 ± 94.63 mg/kg–2,629.71 ± 316.18 mg/kg).

The results obtained were subjected to a PCA to group the different sourdoughs produced based on their similarities or differences in the volatile fraction. Figure 4 shows the result of a previous correlation analysis and Figure 5 presents the PCA biplot obtained. It can be observed that, in general, most compounds had a positive correlation between them, except for the SCFAs (butanoic, pentanoic and hexanoic acids) and heptanoic acid (Figure 4). These compounds were also the ones that characterized the control sourdoughs (Figure 5). Indeed, samples were grouped according to the concentration of Cava lees added to the formulation. The PC1 and PC2 explained 84.40% of the total variability. The first principal component (PC1) explained 75.26% of the samples variances while the second one (PC2) explained 9.15%. All volatile compounds were found on the positive side of PC1, except for butanoic, pentanoic, hexanoic, and heptanoic acids (short- and medium-chain fatty acids), and 2-undecanone (ketone) and (E)-2-Octenal (aldehyde). These compounds were considered to characterize the control and 0.5% Cava lees sourdoughs. On the other hand, it can be observed that the sourdoughs with 1% and 2% Cava lees had the highest concentrations of all volatile compounds, especially esters. PC2 showed a positive correlation with alcohols, aldehydes and ketones, whereas the negative axis contained esters mainly characterizing sourdoughs formulated with 2% lees and acids. 

## 3. Discussion

### 3.1. Optimization of Headspace Solid-Phase Microextraction (HS-SPME) Parameters

As previously stated, although multiple studies have used HS-SPME to extract and analyze the volatile fraction of sourdough [5,6,7,8], there is no common base methodology. Consequently, we conducted an optimization process for the HS-SPME parameters. This included the selection and evaluation of three parameters that influence extraction: heating temperature, equilibrium time, and extraction time. 

Overall, the higher the TN of compounds identified, the greater the TA and concentration of the compounds will likely be [27], as can be observed in Figure 1. Moreover, the increase in heating temperature resulted in a greater composition (TN) and content (TA) of volatilized compounds. This temperature rise could have the ability to facilitate the volatilization of the molecules from the sourdough matrix, improving the vapor pressure and diffusion coefficients of the analytes being absorbed by the fiber coating [27,28]. 

Before extraction, samples are usually equilibrated for a period of time to enable molecules into the headspace, which leads to a potentially greater recovery of the compounds [27,29]. Three sets of time were tested (10 min, 15 min, and 30 min), although there were no statistically significant differences between them.

Lastly, extraction time is of importance since it is the time that compounds need to reach equilibrium between the headspace and the fiber [29]. In fact, longer extraction times can be beneficial, with more analytes occupying more sites on the fiber, but exceeding these times may trigger a desorption of the analytes [27,28]. Additionally, extraction temperature and time are closely related [29], since increasing extraction temperature can accelerate the volatilization of compounds; consequently, extraction time can be reduced. In fact, a three-factor analysis was performed on the results obtained from the optimization to observe any possible interactions between the variables (temperature, equilibrium, and extraction time). It was found that temperature and extraction time presented an interaction, being related to one another in accordance with Garvey et al. (2020) [28]. On that account, the selected HS-SPME parameters for the extraction of volatile compounds in wheat sourdough were 60 °C, 15 min of equilibrium, and 30 min of extraction.

### 3.2. Analysis of Volatile Compounds in Different Sourdough Samples

Volatile compounds in sourdough are developed during the fermentation process, and many come from precursors such as carbohydrates and amino acids. Lipid oxidation also produces aldehydes and ketones from the decomposition of triglycerides and fatty acids. Additionally, LAB also release aroma precursors, such as amino acids that can be transformed into aldehydes or the corresponding alcohols [30]. 

It was observed that, with the addition of Cava lees, there was a greater production of volatile compounds (Table 1), including the products of microbial metabolism. In fact, previous studies have focused on the effect of Cava lees on the growth and survival of LAB, concluding that they have a positive effect on the fermenting microbiota [14,20,21]. Therefore, higher microbial populations in formulations with Cava lees might induce greater concentrations of volatile compounds as a consequence of LAB and yeast fermentation in sourdough. 

#### 3.2.1. Alcohols

Alcohols can be produced by both sugar fermentation (short-chain alcohols) and amino acid metabolism (long-chain alcohols) [30,31], and are usually characterized by green and herbaceous odor notes [32]. In fact, microbial amino acid metabolism may be increased during the back-slopping steps of fermentation as a protection against acidic stress and to maintain the redox balance, transforming peptides and amino acids into higher alcohols [33].

The dominant alcohol quantified was 1-hexanol in the control sourdoughs and in the sourdoughs formulated with Cava lees (Table 2). 1-Hexanol is usually one of the dominant alcohols produced in sourdough [5,7,8], as well as in sparkling wines [31,34,35,36], along with the other alcohols reported in this study, such as 1-pentanol, 1-octanol and 2-ethylhexanol [32,35]. It contributes odors of cut grass, sweetness, resin, flowers and green, and it originates from fermentation and lipid oxidation (linoleic and linolenic acids) [2,5]. Actually, 1-hexanol was also identified in the lees samples (Table 2), which can support the fact that Cava lees seem to retain volatile compounds on their surface during the biological ageing process [15,36]. In addition, it has been reported that heterofermentative bacteria produce a greater quantity of hexanol than homofermentative LAB [5,31,37,38]. In fact, Liu et al. (2020) [38] proposed that facultatively heterofermentative LAB, such as *Lactiplantibacillus plantarum* (formerly *Lactobacillus plantarum*), can produce 1-hexanol via pathways other than the reduction of hexanal and that it can facilitate the production of hexanal, resulting in more substrate to transform into the corresponding alcohol. 

1-Butanol was the dominant alcohol in sourdoughs with 2% Cava lees (249.65 ± 29.86 mg/kg). It can be observed that the mentioned compound increases its concentration by the addition of Cava lees. This higher alcohol has been reported in wine fermentation [39,40] and is also commonly found in sparkling wines [32,35,36,41].

In summary, most of the alcohols identified in the sourdough samples increased their concentration with the addition of Cava lees (Table 1). In addition, the alcohols found in Cava lees increased their concentration in sourdoughs formulated with lees. The highest values were reached at 2% Cava lees (*p* < 0.05), which may be due to higher survival rates among the microorganisms fermenting the sourdough. In fact, it has recently been reported that Cava lees have a growth-promoting effect on different species of LAB in vitro and in sourdough [14,20]. 

#### 3.2.2. Acids

Acids are produced during fermentation throughout the catabolism of long-chain fatty acids [31]. The total concentration of acids was higher with the addition of 2% Cava lees (*p* < 0.05) (Table 1). In general, high concentrations of organic acids exhibit antimicrobial activity, contributing to the extended shelf-life of bread made with sourdough [3]. In this sense, acetic acid in sourdough has a positive effect because, besides improving its sensory properties, it also possesses anti-ripeness and anti-mold activity [5]. In the same manner, it has also been observed in sparkling wine that acids tend to increase in concentration during biological ageing in contact with lees [36]. 

Moreover, there are acids that were not detected in the control sourdoughs that increased in concentration with the addition of lees, as was the case of decanoic acid (Table 3). Decanoic acid has been reported as a major volatile compound found in wine lees surfaces [42]. We identified this compound in Cava lees along with other organic acids, such as acetic, octanoic, dodecanoic, and hexadecanoic acid (Table 3). It can be observed in Table 3 that all of these compounds increased in concentration in the sourdoughs formulated with lees. Since Cava lees can promote the growth and survival of sourdough microbiota [30], this increase may be a consequence of a higher production of microbial metabolites coming from the lees surface, since they are able to retain certain volatile compounds [15,42].

Nevertheless, short-chain fatty acids (SCFAs) (butyric and pentanoic acids) decreased in concentration with the addition of lees. In fact, pentanoic acid was not detected in the sourdoughs with Cava lees, even though both SCFAs are volatiles of fermentation origin that have been reported in wheat sourdough [2,7]. Indeed, butyric acid has been associated with the metabolism of acetic bacteria (such as *Acetobacter cerevisiae*) [7]. Since the sourdoughs produced were fermented spontaneously and analyzed shortly after microbial stabilization, it may be assumed that a greater presence of wild bacterial strains may be conditioned by the addition of lees.

#### 3.2.3. Aldehydes and Ketones

Aldehydes are formed by unsaturated fatty acid decarboxylation as well as lipid oxidation [2,5,7,31]. Hexanal was one of the dominant aldehydes in this study (Table 4). It produces fatty, green, grassy, powerful, and tallow odors and has an odor threshold in water of 4.5–5 ppb [2]. Although lipid oxidation products such as hexanal have been reported several times and in high concentrations in bread crumbs, they generally produce off-flavors [43].

Nonanal, an aldehyde that has also been reported in sparkling wine [32], showed the greatest increment in samples with lees compared to the control, increasing eight times its value when 2% Cava lees (*w*/*w*) were added to the sourdough formulation. Moreover, benzaldehyde was only identified in sourdoughs with 1% and 2% Cava lees. It is an aldehyde commonly found in sparkling wine [31,32] and it has been reported in other foodstuff formulated with wine lees [23]. Nevertheless, the absence of certain aldehydes or their low production may be explained by the ability of heterofermentative LAB to reduce aldehydes to other compounds [37,38].

Regarding ketones, acetoin was the main one in all samples (Table 4). Acetoin is a key aroma in bread formed during fermentation, with a positive correlation with wheat bread; therefore, the higher the concentration, the better the acceptance by consumers [43]. It is characterized by a buttery and creamy odor and comes from the bacterial conversion of citrate into pyruvate, which then results in acetoin in order to equilibrate the redox balance of the cell metabolism [44].

As for 2-undecanone, this ketone has only been reported once in gluten-free hemp-enriched sourdough bread [11], but it has been identified in wine as well [45,46,47]. In this study, 2-undecanone was only found in samples with Cava lees, and it was the only ketone identified in lees (Table 4). This could indicate that it comes from lees, perhaps being attached to their surface during the ageing of the sparkling wine.

#### 3.2.4. Esters

Esters were the most prevalent volatiles in all types of sourdough, especially in samples with Cava lees (Table 1). As a general rule, esters are characterized by a fruity odor, and are a result of the reaction of alcohols (mainly ethanol) and acetyl co-A derivatives of fatty acids [8]. Additionally, ester production is predominantly due to heterofermentative LAB [29,37,38]. In addition, esters are released by the degradation of yeast cells in sparkling wine, which could explain the concentration increase in samples with Cava lees [32,48,49]. Along with other substances, esters improve the flavor characteristics of sourdough bread [3]. Adding Cava lees to sourdough fermentation presents an increment in ester concentration. Therefore, the flavor of the breads produced with these sourdoughs could be more complex.

Some of the esters reported in sourdough with Cava lees were not found in the control (ethyl laurate and decyl acetate). These compounds have previously been reported in wine and sparkling wine, being dependent of the yeast strain as well as the grapes used [32,50,51,52]. So, it can be assumed that these esters originate the Cava lees added to sourdough fermentation. 

Overall, all sourdoughs shared 14 volatile compounds (Figure 6). Moreover, 13 volatiles were identified in both sourdoughs and Cava lees including 2-ethylhexanol, 1-hexanol, 1-pentanol, and 1-octanol (alcohols); hexadecenoic, octanoic, and acetic acid (acids); decanal and nonanal (aldehydes); and ethyl octanoate, hexyl acetate, ethyl decanoate, and ethyl palmitate (esters). Furthermore, ethyl laurate, decyl acetate, 2-undecanone, and decanoic acid were compounds found in the sourdoughs with lees that were also identified in Cava lees. Nevertheless, heptanoic and pentanoic acids were only detected in the control sourdoughs. To summarize, the addition of Cava lees resulted in sourdoughs with a greater diversity of aldehydes and esters, as well as higher concentrations of all chemical families (Table 1). For instance, benzaldehyde, (E)-2-heptanal, (E)-2-octenal (aldehydes), ethyl hexanoate, ethyl laurate, octyl acetate, and decyl acetate (esters) were only produced in sourdoughs with Cava lees.

Lastly, a PCA was performed with the aim to observe the differentiation between the produced sourdoughs (Figure 5). After the analysis, the PCA showed that there were differences between the sourdoughs according to the percentage of Cava lees added to the formulations. It showed that sourdoughs with 1% and 2% Cava lees were described by esters. Oppositely, the control and 0.5% lees sourdoughs were only characterized by short- and medium-chain carboxylic acids. Finally, it is known that there are several factors influencing the volatile characteristics of sparkling wine, such as the grape used, the fermenting yeast, and the terroir [31]. This may also modify the characteristics of the corresponding lees; therefore, further studies should focus on how different lees may impact sourdough and sourdough bread flavor as well as its microbial population and physicochemical characteristics. 

## 4. Materials and Methods

### 4.1. Preparation and Propagation of Sourdoughs

For the sourdough formulation, a commercial wheat flour was used (7230 Buonpane, Molino Quaglia SpA, Padua, Italy) with the following composition (g/100 g): carbohydrates 72.0, fat 1.5, fibre 2.0, protein 11.5, and moisture 15.0.

Sourdoughs were prepared by mixing 100 g of flour and 100 mL of sterile distilled water, without the inoculation of starter culture bacteria or yeasts, and incubated at room temperature for 24 h, following the method described by Martín-Garcia et al. (2022) [29]. Briefly, Cava lees were provided by the winery Freixenet S.A. (Sant Sadurní d’Anoia, Spain) and lyophilized following the method described by Hernández-Macias et al. (2021) [20]. They were added as a percentage of flour weight at different concentrations (0%, 0.5%, 1%, and 2%) to assess their effect on the volatile compounds. Sourdoughs were propagated by backslopping for 8 days and inoculating an aliquot of the previous dough into a new mixture of flour and water. Three different sourdoughs were prepared and analyzed in triplicate.

### 4.2. Optimization of Headspace Solid-Phase Microextraction (HS-SPME) Parameters

The optimization of extraction of volatile compounds was performed using headspace solid-phase microextraction (HS-SPME) and it was carried out using a 2 cm long divinylbenzene/carboxen/polydimethylsiloxane (DVB/CAR/PDMS) fiber supplied by Supelco (Bellefonte, PA, USA). To that end, a control sourdough was produced, and samples of 5 g were prepared. Before extraction, the fiber was conditioned according to the manufacturer’s recommendations. After equilibration at a specified temperature (20, 50, 60 and 75 °C) for a specified time (10, 15 and 30 min), the fiber was exposed to the sample headspace for a specified time (20, 30, 40 and 50 min). A total of 48 runs were analyzed in triplicate for the optimization procedure based on a multilevel factorial design. Once the HS-SPME method was optimized, it was applied to the assessment of the different sourdoughs produced with and without lees. An internal standard (4-methyl-2-pentanol (CAS: 108-11-2, TCI Ltd., Eschborn, Germany), 100 µg/mL) was added (100 µL) for semi-quantification. 

### 4.3. Analysis of Volatile Compounds by Gas Chromatography–Mass Spectrometry (GC-MS)

Chromatographic analysis was carried out in a 6890N Network GC system (Agilent, Palo Alto, CA, USA) coupled to an MS Agilent technologies 5973 Network selective detector (Thermo Fischer Scientific, Waltham, MA, USA). Helium was used as a carrier gas. Separations were accomplished in a DB Wax USN 125-7031 column (30 m × 0.25 mm × 0.25 µm) (Agilent, Palo Alto, CA, USA). A splitless injector suitable for SPME was used. After extraction, the fiber was removed from the headspace vial and inserted directly into the injection port of the GC. The SPME fiber was thermally desorbed for 2.5 min at 260 °C.

The initial temperature was 40 °C for 5 min, and this was subsequently increased at 4 °C/min using the splitless injection mode for 5 min up to 250 °C. GC-MS detection was performed in complete scanning mode (SCAN) in the 40–350 amu mass range with two scans per second. Electron impact mass spectra were recorded at an ionization voltage of 70 eV and an ion source of 280 °C. The volatile concentrations reported were calculated by dividing the peak area of the compounds of interest by the peak area of the internal standard (normalized area). The relative response factor was considered to be 1. Identification was performed by comparison of the mass spectra with the mass spectra library database Wiley 6.0., and retention times with those of pure standards when they were available.

### 4.4. Statistical Analysis

All assays were performed in triplicate and in a randomized run order. The statistical analysis was performed using the Prism 9 version 9.1.2 (225) (GraphPad Software, LLC., San Diego, CA, USA) statistical package. The results are reported as the means ± standard error (SE) for parametric data. A three-factor analysis was conducted on the optimization results. A one-way ANOVA and comparison of the means were conducted using Tukey’s test with a confidence interval of 95%. Significant results were identified with a *p*-value of ≤0.05. Principal component analysis (PCA) was also performed to determine the differences between the sourdoughs.

## 5. Conclusions

After the optimization of the HS-SPME parameters, it was found that the best temperature of extraction was 60 °C, with 15 min of equilibrium and 30 min of extraction for wheat sourdough. Then, when applied to the studied sourdoughs, it was found that acids and esters were the most prevalent compounds quantified, followed by alcohols, aldehydes, and finally ketones. Regarding particular compounds, butyl butyrate, ethyl octanoate, ethyl decanoate, octanoic acid, and 1-hexanol were the most prevalent volatiles quantified.

In general, the addition of Cava lees caused an increase in the concentration of the volatile compounds typically found in sourdough, such as 1-hexanol, acetic acid, hexanal, and ethyl decanoate. Additionally, compounds usually reported in sparkling wines were also identified in sourdough samples formulated with Cava lees, such as 1-butanol, octanoic acid, benzaldehyde, and ethyl hexanoate. Therefore, it can be concluded that Cava lees not only promote the production of sourdough volatile compounds, but they also provide volatiles frequently found in sparkling wines, which supports the fact that lees can retain volatile compounds on their surface. Moreover, the ability of Cava lees to retain odorous volatile compounds could be of great interest for the food and aroma industries that could revalorize and use such by-products, contributing to a circular economy.

## Figures and Tables

**Figure 1 molecules-27-03588-f001:**
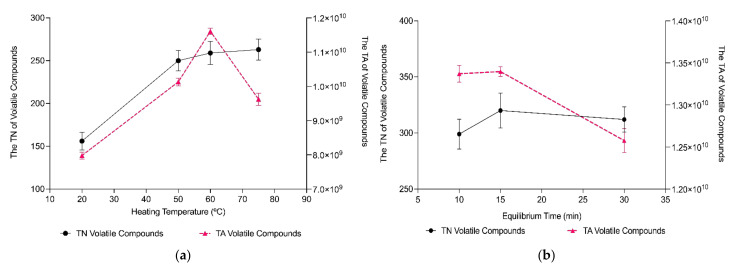
The effect of different extraction parameters of HS-SPME on the total number (TN) and total area (TA) of volatile compounds in sourdough: heating temperature (**a**); equilibrium time (**b**); extraction time (**c**).

**Figure 2 molecules-27-03588-f002:**
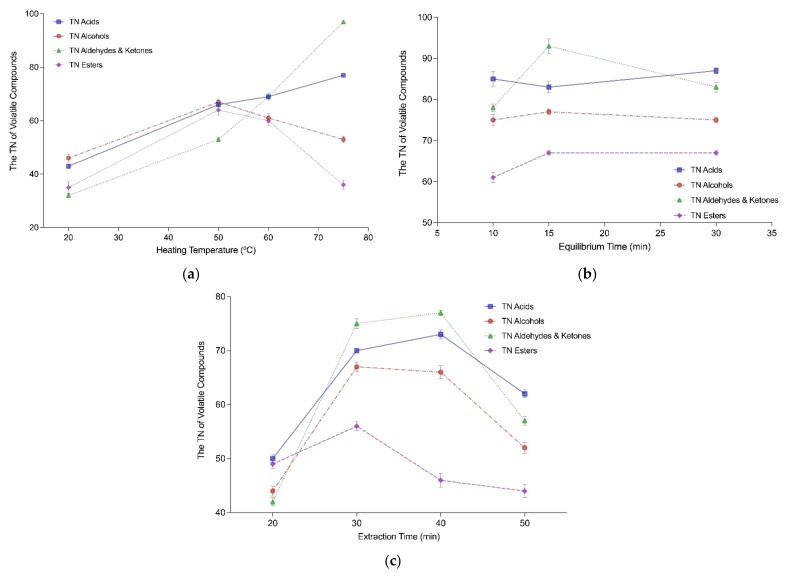
Effect of different extraction parameters of HS-SPME on the total number (TN) of compounds of the different chemical families in sourdough: heating temperature (**a**); equilibrium time (**b**); extraction time (**c**).

**Figure 3 molecules-27-03588-f003:**
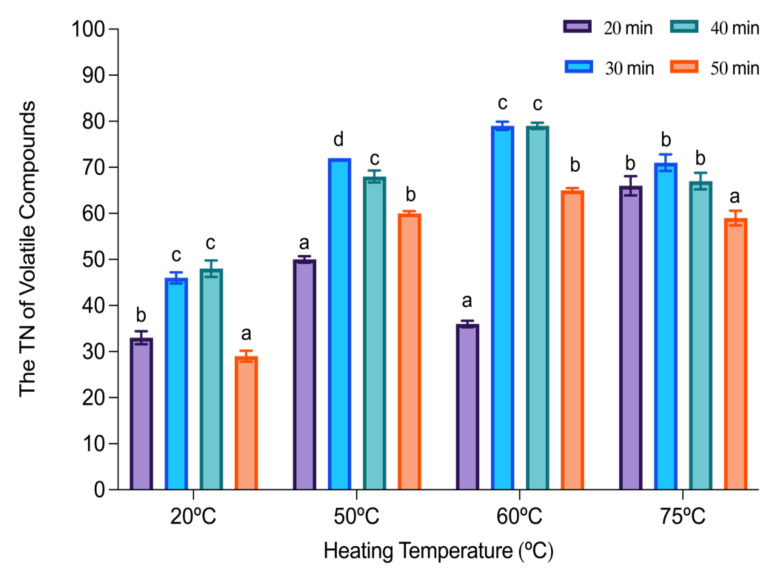
The total number (TN) of volatile compounds regarding heating temperature (°C) and extraction time (min). Different letters denote statistically significant differences (*p* < 0.05) between different times of extraction for each temperature.

**Figure 4 molecules-27-03588-f004:**
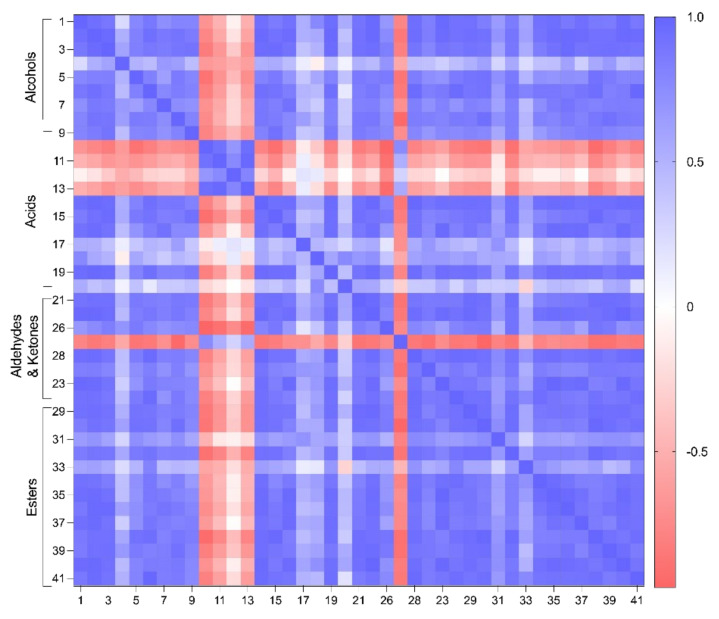
Heatmap of the correlation matrix of the volatile compounds (*p* < 0.05). Numbers in loadings correspond to the volatile compounds identified in sourdoughs: 1–8 alcohols (Table 2); 9–20 acids (Table 3); 21–28 aldehydes and ketones (Table 4); and 29–41 esters (Table 5). Positive correlations are shown in blue; negative correlations in red; absence of correlation in white.

**Figure 5 molecules-27-03588-f005:**
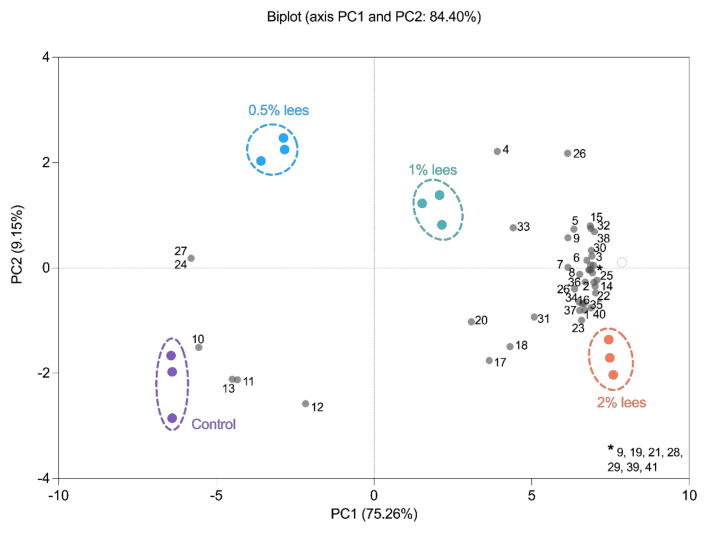
Principal component analysis (PCA) loadings for 41 volatile compounds (grey) and scores for the different sourdoughs at the end of the fermentation period (purple—control; blue—0.5% Cava lees; green—1% Cava lees; and orange—2% Cava lees). Numbers in loadings correspond to the volatile compounds identified in sourdoughs: 1–8 alcohols (Table 2); 9–20 acids (Table 3); 21–28 aldehydes and ketones (Table 4); and 29–41 esters (Table 5).

**Figure 6 molecules-27-03588-f006:**
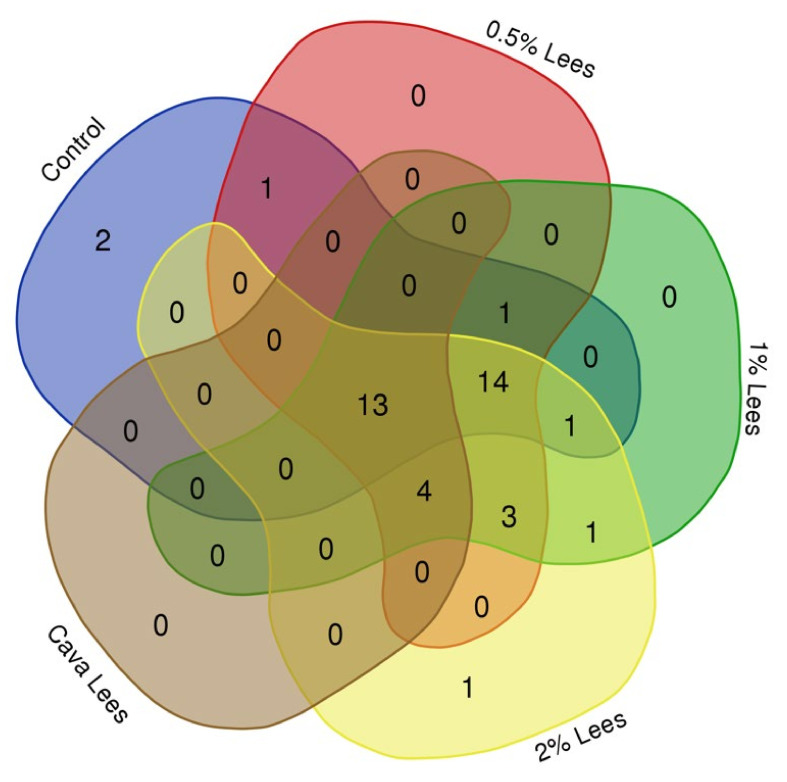
Venn diagram of the volatile compounds shared between the different sourdoughs produced and Cava lees.

**Table 1 molecules-27-03588-t001:** Concentration (mg/kg) of total volatile compounds classified by the chemical family obtained with the optimized HS-SPME method in Cava lees and sourdough.

Family Compound	Cava Lees	Sourdough			
		Control	0.5% Cava Lees	1% Cava Lees	2% Cava Lees
Alcohols	95.14 ± 8.92	158.66 ± 28.81 ^a^	283.87 ± 39.69 ^ab^	505.29 ± 80.88 ^b^	923.39 ± 150.02 ^c^
Acids	143.33 ± 14.45	612.72 ± 39.37 ^a^	386.83 ± 99.92 ^a^	568.27 ± 132.07 ^a^	1008.03 ± 69.33 ^b^
Aldehydes	5.19 ± 1.77	160.44 ± 13.34 ^a^	189.96 ± 12.18 ^a^	165.14 ± 8.61 ^a^	234.31 ± 19.95 ^b^
Ketones	2.01 ± 0.58	11.74 ± 3.95 ^a^	11.59 ± 5.41 ^a^	32.96 ± 7.27 ^b^	44.51 ± 4.19 ^b^
Esters	489.38 ± 60.34	373.08 ± 44.02 ^a^	1136.39 ± 268.77 ^b^	3593.89 ± 737.88 ^c^	7514.18 ± 764.37 ^d^

Values are mean ± standard deviation of triplicates. Significant differences between sourdoughs are indicated by different superscript letters (*p* < 0.05) for each family compound.

**Table 2 molecules-27-03588-t002:** Concentration (mg/kg) of the main alcohols obtained with the optimized extraction parameters in Cava lees and sourdough.

Compound		CAS Num.	Odor ^1^	ODT ^2^	Cava Lees	Sourdough			
						Control	0.5% Cava Lees	1% Cava Lees	2% Cava Lees
1	1-Butanol	71-36-3	medicinal, fruit, wine	500	nd	27.53 ± 9.86 ^a^	58.52 ± 9.06 ^ab^	148.09 ± 36.70 ^b^	390.76 ± 66.00 ^c^
2	1-Pentanol	71-41-0	green, fruit, balsamic	4000	29.74 ± 4.62	6.62 ± 1.73 ^a^	16.04 ± 6.87 ^ab^	41.53 ± 6.72 ^c^	82.42 ± 6.64 ^d^
3	1-Hexanol	111-27-3	sweet, resin, flower	2500	52.76 ± 3.22	76.17 ± 7.93 ^a^	126.41 ± 11.94 ^a^	165.10 ± 21.78 ^ab^	249.65 ± 29.86 ^c^
4	1-Octen-3-ol	2291-86-4	mushroom, earthy	1	nd	20.49 ± 0.96 ^a^	23.28 ± 1.70 ^a^	25.63 ± 2.21 ^a^	24.54 ± 4.30 ^a^
5	1-Heptanol	111-70-6	herb, mushroom, chemical, green	3	nd	17.3 ± 2.88 ^a^	21.91 ± 2.39 ^ab^	28.18 ± 1.56 ^bc^	29.47 ± 2.19 ^cd^
6	1-Octanol	111-87-5	moss, nut, mushroom, chemical	110–130	12.32 ± 1.03	5.12 ± 2.73 ^a^	25.30 ± 2.37 ^ab^	62.66 ± 6.47 ^bc^	100.57 ± 30.93 ^cd^
7	1-Nonanol	143-08-8	fat, green, oily, floral	50	nd	2.60 ± 2.59 ^a^	9.64 ± 3.49 ^ab^	24.63 ± 4.32 ^bc^	34.69 ± 8.16 ^cd^
8	2-Ethylhexanol	104-76-7	citrus, fatty	na	0.32 ± 0.05	2.83 ± 0.13 ^a^	2.77 ± 1.87 ^ab^	9.47 ± 1.12 ^c^	11.29 ± 1.94 ^cd^

^1^ From [25]. ^2^ ODT: Odor Detection Threshold (in water) from [26]. Expressed as mg/kg. Values are mean ± standard deviation of triplicates. Significant differences between sourdoughs are indicated by different superscript letters (*p* < 0.05) for each compound. nd: not detected; na: not available.

**Table 3 molecules-27-03588-t003:** Concentration (mg/kg) of main acids obtained with the optimized extraction parameters in Cava lees and sourdough.

Compound		CAS Num.	Odor ^1^	ODT ^2^	Cava Lees	Sourdough			
						Control	0.5% Cava Lees	1% Cava Lees	2% Cava Lees
9	Acetic acid	64-19-7	pungent, sour	na	32.58 ± 1.24	66.08 ± 9.99 ^a^	132.11 ± 46.11 ^a^	168.78 ± 78.48 ^a^	246.10 ± 14.64 ^ab^
10	Butanoic acid	107-92-6	sweaty, rancid	240	nd	269.41 ± 9.36 ^a^	80.43 ± 14.13 ^b^	29.55 ± 4.26 ^c^	nd
11	Pentanoic acid	109-52-4	-	3000	nd	6.18 ± 2.04	nd	nd	nd
12	Hexanoic acid	142-62-1	-	3000	nd	55.56 ± 2.09 ^ab^	31.60 ± 9.58 ^a^	32.39 ± 3.08 ^a^	41.73 ± 5.97 ^ab^
13	Heptanoic acid	111-14-8	-	3000	nd	8.28 ± 1.23	nd	nd	nd
14	Octanoic acid	124-07-2	oily, rancid	3000	49.45 ± 4.98	8.72 ± 3.18 ^a^	48.40 ± 22.33 ^ab^	135.98 ± 13.91 ^c^	253.58 ± 22.36 ^d^
15	Nonanoic acid	112-05-0	fatty, mild, nutlike	3000	nd	3.03 ± 0.80 ^a^	17.78 ± 3.17 ^b^	28.21 ± 7.51 ^b^	40.99 ± 1.93 ^c^
16	Decanoic acid	334-48-5	sour, fatty	10,000	31.68 ± 5.84	nd	23.27 ± 1.14 ^a^	85.70 ± 11.43 ^b^	209.27 ± 4.98 ^c^
17	Benzoic acid	1863-63-4	-	na	nd	1.33 ± 0.05 ^a^	0.61 ± 0.33 ^b^	1.50 ± 0.02 ^a^	1.56 ± 0.04 ^a^
18	Tetradecanoic acid	544-63-8	waxy, oily, faint	10,000	nd	5.55 ± 0.17 ^a^	nd	11.37 ± 1.36 ^b^	29.36 ± 4.98 ^c^
19	Hexadecanoic acid	57-10-3	-	10,000	29.62 ± 2.39	8.57 ± 5.82 ^a^	52.63 ± 3.13 ^bc^	74.79 ± 12.02 ^bc^	164.35 ± 10.79 ^d^
20	Octadecanoic acid	57-11-4	-	20,000	nd	nd	nd	nd	21.09 ± 3.64

^1^ From [25]. ^2^ ODT: Odor Detection Threshold (in water) from [26]. Expressed as mg/kg. Values are mean ± standard deviation of triplicates. Significant differences between sourdoughs are indicated by different superscript letters (*p* < 0.05) for each compound. nd: not detected; na: not available.

**Table 4 molecules-27-03588-t004:** Concentration (mg/kg) of main aldehydes and ketones obtained with the optimized extraction parameters in Cava lees and sourdough.

Compound		CAS Num.	Odor ^1^	ODT ^2^	Cava Lees	Sourdough			
						Control	0.5% Cava Lees	1% Cava Lees	2% Cava Lees
21	Hexanal	66-25-1	fatty, green, grassy	4.5–5	nd	45.32 ± 3.11 ^a^	50.85 ± 1.73 ^ab^	60.55 ± 2.55 ^bc^	69.19 ± 5.00 ^bc^
22	Acetoin	513-86-0	butter, cream	800	nd	11.74 ± 3.95 ^a^	8.28 ± 4.44 ^a^	24.09 ± 3.99 ^b^	29.29 ± 1.59 ^b^
23	Benzaldehyde	100-52-7	cherry, candy	350–3500	nd	nd	nd	0.55 ± 0.08	2.59 ± 0.34
24	2-Undecanone	112-12-9	citrus, rose, iris	7	2.01 ± 0.58	nd	3.31 ± 0.98 ^a^	8.86 ± 3.28 ^a^	14.86 ± 2.60 ^b^
25	Nonanal	124-19-6	piney, floral, citrusy, fat	1	3.21 ± 1.10	9.94 ± 2.65 ^a^	20.08 ± 1.85 ^ab^	46.65 ± 3.08 ^c^	88.28 ± 9.06 ^d^
26	(E)-2-Heptenal	18829-55-5	green, sweet, fresh, fruity, apple	13	nd	nd	5.13 ± 0.72 ^a^	6.71 ± 0.55 ^a^	7.41 ± 0.92 ^ab^
27	(E)-2-Octenal	2548-87-0	green, nut, fat, leaf, walnut	3	nd	72.62 ± 3.32 ^a^	75.18 ± 5.44 ^a^	nd	nd
28	Decanal	112-31-2	beefy, musty, marine, cucumber	0.1–2	1.98 ± 0.67	32.56 ± 4.27 ^a^	38.72 ± 2.43 ^ab^	50.68 ± 2.35 ^c^	66.84 ± 4.63 ^d^

^1^ From [25]. ^2^ ODT: Odor Detection Threshold (in water) from [26]. Expressed as mg/kg. Values are mean ± standard deviation of triplicates. Significant differences between sourdoughs are indicated by different superscript letters (*p* < 0.05) for each compound. nd: not detected.

**Table 5 molecules-27-03588-t005:** Concentration (mg/kg) of major esters obtained with the optimized extraction parameters in Cava lees and sourdough.

Compound		CAS Num.	Odor ^1^	ODT ^2^	Cava Lees	Sourdough			
						Control	0.5% Cava Lees	1% Cava Lees	2% Cava Lees
29	Butyl acetate	123-86-4	sweet, ripe, fruity, green	66	nd	31.01 ± 5.70 ^a^	43.39 ± 4.03 ^b^	55.62 ± 3.84 ^b^	73.55 ± 4.90 ^c^
30	Butyl butyrate	109-21-7	fruity, sweet	100	nd	96.76 ± 3.36 ^a^	222.68 ± 41.42 ^b^	597.88 ± 36.26 ^c^	689.04 ± 12.35 ^d^
31	Butyl hexanoate	626-82-4	fruity, winey, berry, green	700	nd	67.34 ± 10.18 ^a^	63.24 ± 3.82 ^a^	76.95 ± 3.75 ^a^	79.26 ± 2.87 ^a^
32	Butyl benzoate	136-60-7	amber, balsamic, fruity	na	nd	56.68 ± 2.10 ^a^	69.93 ± 6.29 ^b^	82.88 ± 3.76 ^c^	91.73 ± 2.23 ^c^
33	Ethyl hexanoate	123-66-0	fruity	70–84	nd	nd	31.66 ± 5.89 ^a^	49.81 ± 9.52 ^a^	100.75 ± 14.96 ^b^
34	Ethyl octanoate	106-32-1	fruity, floral	na	139.63 ± 12.67	6.98 ± 1.65 ^a^	170.91 ± 94.63 ^a^	1503.39 ± 275.45 ^b^	2,629.71 ± 316.18 ^c^
35	Ethyl decanoate	110-38-3	sweet, oily, nutlike	na	250.07 ± 17.95	80.07 ± 15.41 ^a^	404.93 ± 79.33 ^ab^	930.25 ± 351.91 ^b^	3,330.26 ± 314.82 ^c^
36	Ethyl laurate	106-33-2	sweet, waxy, creamy, floral	na	41.33 ± 13.46	nd	32.54 ± 15.66 ^a^	70.19 ± 16.67 ^a^	189.99 ± 48.46 ^b^
37	Ethyl palmitate	628-97-7	waxy, fruity, creamy, vanilla, balsamic	>2000	13.59 ± 1.58	4.75 ± 1.58 ^a^	5.14 ± 2.49 ^a^	21.33 ± 13.77 ^a^	60.40 ± 6.59 ^b^
38	Hexyl acetate	142-92-7	sweet, fruity, herb	2	24.79 ± 8.45	3.99 ± 1.24 ^a^	42.61 ± 2.97 ^b^	88.99 ± 8.01 ^c^	115.10 ± 7.22 ^d^
39	Hexyl butyrate	2639-63-6	green, fruity, vegetable, waxy	250	nd	25.50 ± 2.80 ^a^	41.99 ± 8.76 ^a^	79.49 ± 8.90 ^b^	104.75 ± 19.55 ^bc^
40	Octyl acetate	112-14-1	fruity, fatty	12	nd	nd	1.75 ± 0.80 ^a^	19.87 ± 4.41 ^b^	49.70 ± 6.24 ^c^
41	Decyl acetate	112-17-4	sweet, fatty, fruity	na	19.97 ± 6.23	nd	5.62 ± 2.67 ^a^	17.25 ± 1.63 ^b^	29.94 ± 8.00 ^c^

^1^ From [25]. ^2^ ODT: Odor Detection Threshold (in water) from [26]. Expressed as mg/kg. Values are mean ± standard deviation of triplicates. Significant differences between sourdoughs are indicated by different superscript letters (*p* < 0.05) for each compound. nd: not detected; na: not available.

## Data Availability

Not applicable.

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
