# Peer review of "Changes in the Volatile Profile of Wheat Sourdough Produced with the Addition of Cava Lees"

_molecules, 2022, doi:10.3390/molecules27113588_

Round 1
Reviewer 1 Report
Comments to Authors,
The manuscript "Changes in the Volatile Profile of Wheat Sourdough Produced with the Addition of a Winery By-Product" is suitable for publication in Molecules; however, it has to be improved in some aspects, considering the following general and specific comments.
General comments:
In general, the study is interesting and significant in terms of the methodological contribution for this type of analysis and determination of volatile compounds produced by adding a by-product of wine processing to wheat sourdough; however, it would have been interesting to carry out some other type of design or statistical analysis, for example, to evaluate the combined effect of temperature, extraction time and equilibrium time through a three-factor analysis. In addition to the above, the manuscript should be improved in relation to the following specific comments.
Specific comments:
Line 40: The authors should indicate in parentheses the meaning of the initials of the HS-SPME technique.
Lines 84-95: Separate the quantity (magnitude) from the unit of measure (°C) in relation to the temperature quantities and revise or correct this throughout the manuscript.
Figure 1a and b: Indicate in both Figures on the ordinate axis (“y”, to the right) the evaluation of “The TA of volatile compounds” as indicated in Figure 1c.
Figure 3: Indicate the letters of statistical significance.
Line 95: Indicate the probability value (p < 0.05, or p £ 0.05) where it is indicated that “there was no significant difference”.
Line 135, It should read: 60 °C.
Lines 401 and 402: The authors should indicate why the temperatures of 20, 50, 60 and 75 °C were used or what the reason for specifically using these temperatures is and why other temperatures were not used for the extraction conditions by HS-SPME.
In addition to the above, if the intention was to evaluate three independent variables (temperature, extraction time and equilibrium time) with different levels each on the efficiency or optimization of the parameters of the HS-SPME technique: Why the authors did not perform a tri-factorial analysis in order to obtain a possible interaction between the variables analyzed?
Author Response
Thank you for your comments and suggestions in order to improve the paper.
The three-factor analysis was added in 4.4. Statistical analysis section (468-469). And the results obtained were added in the discussion section (274-278).
Specific comments:
Line 40: The authors should indicate in parentheses the meaning of the initials of the HS-SPME technique.
The meaning of HS-SPME was added to the text (Line 41).
Lines 84-95: Separate the quantity (magnitude) from the unit of measure (°C) in relation to the temperature quantities and revise or correct this throughout the manuscript.
The quantity and the unit of measure were separated throughout all the manuscript.
Figure 1a and b: Indicate in both Figures on the ordinate axis (“y”, to the right) the evaluation of “The TA of volatile compounds” as indicated in Figure 1c.
The axis of Figure 1a and 1b were corrected.
Figure 3: Indicate the letters of statistical significance.
The letters of statistical significance were added.
Line 95: Indicate the probability value (p < 0.05, or p £ 0.05) where it is indicated that “there was no significant difference”.
The probability value (p > 0.05) was added (Line 98).
Line 135, It should read: 60 °C.
It was corrected (Line 139).
Lines 401 and 402: The authors should indicate why the temperatures of 20, 50, 60 and 75 °C were used or what the reason for specifically using these temperatures is and why other temperatures were not used for the extraction conditions by HS-SPME.
The temperatures used by the method optimization were fixed according to the previous experience of our research group with SPME analysis. Also, the literature about bakery products consulted used these range of temperatures. Less than 20 ºC was not possible in our lab, while more than 75 ºC can led to artifact formation.
In addition to the above, if the intention was to evaluate three independent variables (temperature, extraction time and equilibrium time) with different levels each on the efficiency or optimization of the parameters of the HS-SPME technique: Why the authors did not perform a tri-factorial analysis in order to obtain a possible interaction between the variables analyzed?
The trifactorial analysis was performed and added to Statistical analysis and results and discussion. According to the results obtained, it was found that time and temperature were related and interact between them. It was added to the text (Lines 274-278 and 468-469).
Reviewer 2 Report
The authors submitted a manuscript focusing on two key topics:
--the valorization of wine industry by-products such as lees to improve the volatile profile of sourdough
--the optimization of the extraction method of volatiles from the studied matrix.
While the topics are interesting, there are revisions to report:
INTRODUCTION:
The scope of the paper is not well described, improve this part;
Materials and methods:
Line 398: The authors proposed optimizing the parameters of the HS-SPME procedure, why did they use DVB-CAR-PDMS fiber? Generally, the choice of fiber is also among the extraction optimization parameters;
Also, why were other parameters, such as sample volume and salt salting-out effect, not considered?
Results:
Figure 4 is not well presented in terms of the results shown and their discussion.
Conclusions:
There is no reference in the conclusion paragraph to the results of HS-SPME parameter optimization, which is part of the manuscript.
Author Response
Thank you for your comments and suggestions. They can be considered in order to improve the manuscript. The list of changes and the responses were added below.
INTRODUCTION:
The scope of the paper is not well described, improve this part;
The introduction was improved in order to better describe the scope of the paper. Also, the optimization of HS-SPME method was added as an objective of the paper (Lines 51-54, 69).
Materials and methods:
Line 398: The authors proposed optimizing the parameters of the HS-SPME procedure, why did they use DVB-CAR-PDMS fiber? Generally, the choice of fiber is also among the extraction optimization parameters;
The use of DVB-CAR-PDMS fiber was based on our previous experience in HS-SPME. The use of a longer fiber and the combination of three different polymers results in a wider number of compounds and more diversity in the chemical families than when the extraction is performed with one polymer.
Also, why were other parameters, such as sample volume and salt salting-out effect, not considered?
The sample volume was not considered based on our previous experience. While the salting out effect was tried but the results were not better compared with those without salt.
Results:
Figure 4 is not well presented in terms of the results shown and their discussion.
Figure 4 was better explained in results and in discussion in order to improve this part of the paper (Lines 224-228).
Conclusions:
There is no reference in the conclusion paragraph to the results of HS-SPME parameter optimization, which is part of the manuscript.
The results of HS-SPME optimization were added to conclusion section (Lines 474-476).
Reviewer 3 Report
Comments (molecules-1742355):
This manuscript titled “Changes in the Volatile Profile of Wheat Sourdough Produced with the Addition of a Winery By-Product” evaluate the impact of Cava lees on sourdough volatile compounds by HS-SPME-GC-MS. It is an interesting idea for bread production. The experiments are coherent, and the results are well presented. However, some problems have to be addressed for authors’ consideration.
- “Cava lees” might be more appropriate “word” compared to “a Winery By-Product” of the title.
- Line 20. What’s “re-valorize” stands for?
- Authors should reorganize the section Abstract, for too many background description while little results information.
- Why authors used Cava lees to improve volatile profile of wheat sourdough produced? The reasons should be supplemented into the section Introduction.
- To illustrate changes of volatile profiles before and after adding Cava lees clearly, a comparison diagram should be drawn in the section Discussion.
- Why butanoic acid decreased after adding Cava lees (see table 3)? Authors should make some discussions.
- It’s interesting to find same volatiles as sparkling wine does after addition of Cava lees to sourdough, so authors should make more descriptions.
Author Response
Thank you for your comments and suggestions in order to improve the paper.
“Cava lees” might be more appropriate “word” compared to “a Winery By-Product” of the title.
The title was changed according the suggestion of the reviewer.
Line 20. What’s “re-valorize” stands for?
Revalorize explore the possibility of reusing by-products of the food industry in the production of main products as bread. During the production of Spanish sparkling wine, approximately 300 T/year of lees (by-product) are generated. Currently, they are only destined to distillery without added value regardless of their composition rich in fiber and antioxidants.
Authors should reorganize the section Abstract, for too many background description while little results information.
The reviewer is correct, the abstract was reorganized in order to introduce more results information.
Why authors used Cava lees to improve volatile profile of wheat sourdough produced? The reasons should be supplemented into the section Introduction.
The reasons of use of cava lees were added to the introduction (Lines 51-54).
To illustrate changes of volatile profiles before and after adding Cava lees clearly, a comparison diagram should be drawn in the section Discussion.
According to the suggestion of the reviewer it was introduced in Discussion a Venn diagram in order to compare the volatile profiles before and after the addition of Cava lees (Figure 6).
Why butanoic acid decreased after adding Cava lees (see table 3)? Authors should make some discussions.
A discussion about the butanoic acid and other short chain fatty acids were added (Lines 345-352).
It’s interesting to find same volatiles as sparkling wine does after addition of Cava lees to sourdough, so authors should make more descriptions.
According to the suggestion of the reviewer more descriptions were added in order to discuss the volatiles of sparkling wine after the addition of Cava lees in sourdough in the discussion section.
Round 2
Reviewer 2 Report
Dear Authors,
I accept the revisions made by the authors and there are no other revisions to report
Author Response
Thank you very much for to revise the manuscript.
Reviewer 3 Report
Comments (molecules-1742355):
Authors have revised their manuscript, and the quality of this revised manuscript has been improved a lot. However, minor modifications should be made before ready for acceptance.
1.Line 74-75, References need to be supplemented.
2. Line 401. Authors should make a summary on which kinds of volatile were produced after addition of Cava Lees.
Author Response
Thank you for to revise the manuscript. The changes were performed:
Line 74-75, the references were added.
Lines 402-406, a summary with the volatiles produced after addition of cava lees was added.